# Night-Time Heart Rate Variability during an Expedition to Mt Everest: A Case Report

**DOI:** 10.3390/sports11020048

**Published:** 2023-02-20

**Authors:** Konstantinos Mantzios, Aggelos Pappas, Georgios-Ioannis Tsianos, Andreas D. Flouris

**Affiliations:** 1FAME Laboratory, Department of Physical Education and Sport Science, University of Thessaly, 42100 Trikala, Greece; 2Department of Physical Education and Sport Science, University of Thessaly, 42100 Trikala, Greece

**Keywords:** altitude, HRV, sleep, performance

## Abstract

Mt Everest has been gaining popularity from casual hiking athletes, climbers, and ultra-endurance marathon runners. However, living and sleeping at altitude increases the risk of injury and illness. This is because travel to high altitudes adversely affects human physiology and performance, with unfavourable changes in body composition, exercise capacity, and mental function. This is a case report of a climber who reached the summit of Mt Everest from the north side. During his 40-day expedition, we collected sleep quality data and night-time heart rate variability. During the night inside the tent, the air temperature ranged from −12.9 to 1.8 °C (−5.8 ± 4.9 °C) and the relative humidity ranged from 26.1 to 78.9% (50.7 ± 16.9%). Awake time was 17.1 ± 6.0% of every sleep-time hour and increased with altitude (r = 0.42). Sleep time (r = −0.51) and subjective quality (r = 0.89) deteriorated with altitude. Resting heart rate increased (r = 0.70) and oxygen saturation decreased (r = −0.94) with altitude. The mean NN, RMSSD, total power, LF/HF, and SD1 and SD2 were computed using the NN time series. Altitude reduced the mean ΝΝ (r = −0.73), RMSSD (r = −0.31), total power (r = −0.60), LF/HF ratio (r = −0.40), SD1 (r = −0.31), and SD2 (r = −0.70). In conclusion, this case report shows that sleeping at high altitudes above 5500 m results in progressively reduced HRV, increased awakenings, as well as deteriorated sleep duration and subjective sleep quality. These findings provide further insight into the effects of high altitude on cardiac autonomic function and sleep quality and may have implications for individuals who frequently spend time at high altitudes, such as climbers.

## 1. Introduction

Year on year, more than 800 climbers attempt to reach the highest peak on the planet: the summit of Mt Everest. In recent decades, Mt Everest has been gaining popularity from casual hiking athletes, climbers, and ultra-endurance marathon runners [1]. Between 1990 and 2005, there were 2200 individuals who attempted the summit for the first time. However, this number has risen dramatically in recent years, with over 3600 first-time climbers attempting the summit between 2006 and 2019 [2]. However, living and sleeping at altitude increases the risk of injury and illness [1,3]. More than 300 people are injured annually while attempting to climb Mt Everest [4]. The mortality rate among climbers at altitudes higher than the Base Camp is 1.3%, and the majority of these deaths (82.3%) occur on or after the summit attempt [5]. This rate ranges from 10 to 12.6 deaths per 100 mountaineers climbing above 6000 m in the Himalayas [6]. The cause of these grim statistics is that travel to high altitudes adversely affects human physiology and performance [7,8], with unfavourable changes in body composition, exercise capacity, and mental function [8,9]. It also leads to symptoms such as fatigue, headaches, and difficulty concentrating, as the body has to work harder to perform the same tasks due to hypoxemia [10]. The limited available evidence suggests that acute exposure to high altitude significantly increases the activity of the sympathetic nervous system [11]. This is accompanied by a decrease in heart rate variability (HRV) [12]. HRV is a widely recognized noninvasive method for evaluating the balance between the parasympathetic and sympathetic nervous systems in the regulation of cardiac function. The ability to assess HRV in individuals has significant clinical implications, as it can provide insights into an individual’s cardiovascular health and risk for disease [13].

Night-time HRV is associated with sleep quality because sleep in healthy individuals is characterized by attenuated sympathetic and increased parasympathetic nervous activity [14,15]. As such, sleep problems are linked with autonomic imbalance [15,16]. This ability of HRV to assess sleep quality can be of value for climbers since the ascent to high altitudes leads to reduced sleep time and subjective sleep quality, as well as increased night-time awakenings [8,9,17]. This is due to the decrease in oxygen levels at high altitude, which can cause respiratory disturbances during sleep. Compared to sea level, sleep at high altitude is associated with decreased slow-wave sleep and REM sleep, sleep efficiency, and total sleep time, and increased waking time [8]. A previous study showed that the ascent to Everest North Base Camp (5180 m) decreased vagal tone and increased sympathetic activity, a phenomenon that was reversed with acclimatization [17], further supporting the disruption of normal HRV during the ascent to high altitude. An autonomic imbalance during sleep, characterized by increased sympathetic activation, may contribute to the development of acute mountain sickness and other negative health outcomes [11]. However, autonomic nervous system fluctuations have not been studied at altitudes beyond 5500 m, which is where the body’s physiology is significantly challenged [18,19]. In this case study, we aimed to explore night-time HRV fluctuations and oxygen saturation at altitudes beyond 5500 m during an expedition to Mt Everest.

## 2. Materials and Methods

The participant of this study was an experienced male climber and ultra-endurance athlete (age 44 years; height 183 cm; body mass 85 kg). He had previously climbed Mt Everest from the north side. Additionally, he had completed multiple extreme ultra-endurance events including the Marathon des Sables run in the Sahara Desert, a race that covers more than 250 km of harsh terrain in extreme heat [20]. He had also swum the English Channel (34 km of cold, open water), and had swum non-stop across the Aegean Sea: a distance of 101 km.

Thirty days before the ascent to Mt Everest, the athlete followed a 4-week altitude acclimation protocol at the FAME Laboratory (Trikala, Greece). The daily protocol was designed to simulate the physical demands of a high-altitude environment and included a combination of running and cycling that were performed over a period of 4 h. The daily protocol included 2 h of running in normobaric normoxia and 2 h of cycling in normobaric hypoxia. This type of hypoxia simulates altitudes between 5000 and 6000 m. The hypoxia environment was achieved using the Everest Summit II (Hypoxico Inc, Gardiner, NY, USA).

The outline of the climber’s ascent profile is illustrated in Figure 1. In brief, he remained at Base Camp (BC) for 6 days (days 6–11) and during this time he performed daily aerobic exercise (stationary running and stepping) for approximately 1 h every day. During days 12–13, he climbed to 6340 m (Advanced Base Camp) where he spent three days and then returned to BC (day 17). He had planned to continue moving between 5000 and 6500 m until weather would permit him to ascend to the summit. Alas, four days after returning to BC (on day 21), he fell while descending from a training climb and injured his finger (Figure 1). He sustained an extensive laceration to the distal palmar aspect of his right middle finger. The injury required 10 sutures and the procedure was undertaken at BC. On day 23, he climbed again to 6340 m, but the hypoxic conditions caused a significant deterioration of his injury, which forced him to descend to Tingri (4348 m) and visit the local hospital for further assessment and treatment with several antibiotic regimes and analgesia. Seven days later, on day 32, he trekked back to the BC. Seven days thereafter (day 39), he successfully reached the summit from the north side. Then, he descended to Camp 3 (8300 m; day 40) and proceeded to lower altitudes.

During the expedition, a custom-built GPS-logger was used to track the location of the climber. This device was developed using an Arduino microcontroller and was equipped with a NEO-6M GPS antenna (Somerville, MA, USA). The GPS-logger was essential in recording the climber’s movements throughout the expedition, providing accurate location data. The device was powered by a 9-volt lithium battery that was specially designed with extra protection for cold temperatures, ensuring that the GPS-logger would function correctly even in the harsh conditions of the expedition. In addition, a small, portable weather station (Kestrel Drop D3, Nielsen-Kellerman Co., Boothwyn, PA, USA) was used to record weather data (air temperature and relative humidity) at various locations and altitudes throughout the expedition. Additionally, oxygen saturation and resting heart rate were collected every morning upon waking up (NONIN GO2 Achieve 9570 finger pulse oximeter, Nonin Medical Inc., Plymouth, MN, USA). During the 40-day expedition, we employed a specific protocol for collecting data on night-time HRV. For 12 nights, the climber wore a modified heart rate monitor (Polar Team 2, Polar Electro Oy, Kempele, Finland) while he slept. This device was powered by a specially designed 3.7 V and 2000 mAh lithium battery with dimensions 6.7 × 44 × 63 mm that was equipped with extra protection to withstand cold temperatures. This allowed us to gather continuous HRV data throughout the night, providing a comprehensive understanding of the climber’s autonomic nervous system activity during sleep. This method was chosen for its non-invasive nature, its ability to capture long-term patterns in HRV, and the device’s ability to operate in extremely cold temperatures.

In the collected HRV series, we removed observations that did not fit within a set range from our dataset and replaced them with linear interpolation to maintain the continuity of the signal. This important preprocessing step helps cleaning up the data and removing noise that may be present due to movement, waking, or coughing [13]. The databased cleaning and analysis of the beat-by-beat series of RR intervals for each entire night was performed with the HRV-library written in Python (https://pypi.org/project/hrv-analysis/ (accessed on 16 August 2021)) using a 5-min window analysis with 2.5 min time step throughout the night-time for every different night. Using the normal-to-normal (NN) time series, we computed the mean NN, the root mean square of successive differences between normal heartbeats (RMSSD), the total power, the low-frequency to high-frequency (in normalized units) ratio (LF/HF), as well as the Poincaré SD1 and SD2 and their ratio (SD1/SD2) [21,22]. In long-term recordings, such as the ones performed in our study, the RMSSD and Poincaré SD1 reflect primarily parasympathetic drive, while the total power, LF/HF, Poincaré SD2, and Poincaré SD1/SD2 are sensitive to both parasympathetic and sympathetic tones [13,22,23]. Finally, after the expedition, we analyzed the climber’s diary and had an extended discussion with him to extract information on sleep quality based on the Stanford Sleepiness Scale (SSS) [24]. Due to the small sample size and non-normal data distribution, we used the Spearman correlation coefficient to detect potential associations between altitude and oxygen saturation, resting heart rate, and HRV parameters.

## 3. Results

During the night inside the tent, the air temperature ranged from −12.9 to 1.8 °C (−5.8 ± 4.9 °C) and the relative humidity ranged from 26.1 to 78.9% (50.7 ± 16.9%). Awakenings were very frequent during the recorded sleeping patterns. For every hour of night-time sleep, the average awake time ranged from 9 to 26.5% (17.1 ± 6.0%) and was linearly associated with altitude (r = 0.42). The heart rate data recorded during awakening were not included in the HRV analysis. We found that the altitude level was positively correlated with the resting heart rate (r = 0.70) and negatively correlated with oxygen saturation (r = −0.94) (Figure 1). After removing all of the awakening time, the average time used for the HRV analysis was 226.8 ± 97.1 min ranging from 117.5 to 455.0 min and was negatively associated with altitude (r = −0.51). Additionally, altitude reduced the mean ΝΝ during sleep (r = −0.73; Figure 2), which was 813.0 ± 42.3 ms at 5400 m (BC; Day 6) compared to 948.2 ± 69.4 ms at 4348 m (Tingri; Day 29). The same was also observed for the other parameters of HRV [RMSSD (r = −0.31); total power (r = −0.60); LF/HF ratio (r = −0.40); Poincaré SD1 (r = −0.31); Poincaré SD2 (r = −0.70); Poincaré SD1/SD2 (r = −0.24)]. Subjective sleep quality deteriorated while ascending at higher altitudes (r = 0.89; higher SSS scores indicate lower subjective sleep quality), being high (1.2 ± 0.4 in the SSS, indicating feeling active, vital, alert, or wide awake during the day) at altitudes 4348–5440 m, moderate (2.2 ± 0.4 in the SSS, indicating functioning at high levels, but not fully alert during the day) at 6340 m, and low (4.0 ± 2.1 in the SSS, indicating being somewhat foggy, or let down during the day) at altitudes 7050–8300 m. Acclimatization to altitude at 5400 m (BC; days 7–10) and at 6340 m (days 13–15 and 34–37) increased the mean NN (average change: 76.44 ± 17.72 ms). This also correlated with subjective sleep quality, which improved during habituation at specific altitudes (i.e., SSS scores were higher on the first day at a given altitude, compared to subsequent days at the same altitude).

## 4. Discussion

This case report presents the first evidence of night-time HRV fluctuations at altitudes beyond 5500 m during an expedition to Mt Everest [in our case, Advanced Base Camp (6340 m), the North Col (7020 m), and Camp 3 (8300 m)]. To our knowledge, this is the first study to directly observe the autonomic nervous system activity at this altitude, filling a significant gap in the literature, as previous studies were either conducted in simulated conditions or on animal subjects [11]. The findings of our case study are valuable, as they provide new insight into the human response to high altitude, and the expected differences in autonomic nervous system activity above 5500 m. A decrease in HRV is linked to a higher likelihood of experiencing acute mountain sickness [25]. Our climber experienced decreased HRV during sleep at these high altitudes, and this was associated with reduced sleep duration and lower subjective sleep quality. These findings confirm previous studies assessing night-time HRV, which reported that the rapid ascent to altitudes up to 5500 m causes sleep apneas and increased sympathetic drive [17]. The latter is triggered by peripheral chemoreceptor stimulation aiming to offset the attenuated arterial oxygen content [17].

Within 3–4 days of acclimatization to altitude at 5400 m (BC) and at 6340 m, we recorded favourable adaptations in HRV and subjective sleep quality. Based on previous reports, acclimatization at 5180 m increases the ventilatory response to hypoxemia [26] and the time domain of the HRV [17]. Our results support this conclusion, as we observed adaptation patterns in the NN, but not in other HRV domains. During the first days of acclimatization at 5400 m (BC), we found an improvement (~4%) in oxygen saturation. This pattern was not systematically observed during the remaining phases of the expedition, which may be explained by the disruption in the pre-planned 1-month acclimatization protocol caused by the injury and the need to descend to a lower altitude. Nevertheless, it is logical to suspect that improvements in sleep quality and HRV time domain measures were caused by favorable adaptations in oxygen delivery. It would have been interesting to confirm this notion with added analyses using daytime HRV data. Unfortunately, our study design did not allow exploring daytime HRV adaptations.

When reading our results for LF/HF, it is important to consider that this parameter is influenced by breathing rate, which was not assessed/controlled in our study. As such, we recommend caution when interpreting our results for the LF/HF ratio as a measure of parasympathetic and sympathetic tones. To address this, we also calculated the Poincaré SD1/SD2 ratio which is not influenced by the respiratory rate. The results of Poincaré SD1/SD2 showed similar, yet less pronounced, effects to those of LF/HF.

## 5. Conclusions

In conclusion, our case report presented here demonstrates that sleeping at high altitudes above 5500 m results in progressively reduced HRV, increased awakenings, as well as deteriorated sleep duration and subjective sleep quality. These findings provide further insight into the effects of high altitude on cardiac autonomic function and sleep quality and may have implications for individuals who frequently spend time at high altitudes, such as climbers.

## Figures and Tables

**Figure 1 sports-11-00048-f001:**
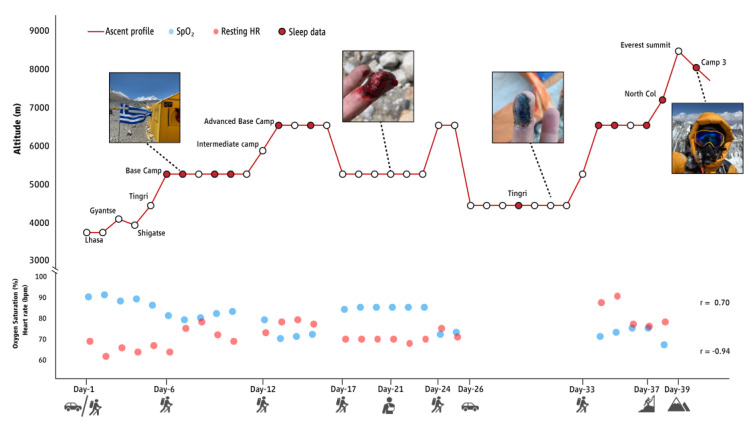
GPS tracking of the 40-day Mt Everest expedition and results for morning oxygen saturation (SpO_2_) and resting heart rate (HR). Circles indicate nights during the expedition. Red circles indicate assessment of heart rate variability during night-time sleep. Correlation coefficients (r) indicate associations with altitude.

**Figure 2 sports-11-00048-f002:**
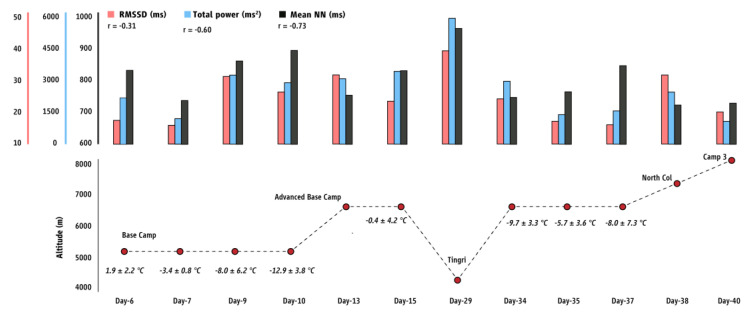
Results for the root mean square of successive differences between normal heartbeats (RMSSD), the total power, and the mean normal-to-normal (NN) beats during night-time sleep at different altitudes. Correlation coefficients (r) indicate associations with altitude. Numbers in italics indicate mean ± SD ambient temperature in the tent during night-time sleep.

## Data Availability

The data that support the findings of this study are available from the corresponding author, upon reasonable request.

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
