# Peer review of "Night-Time Heart Rate Variability during an Expedition to Mt Everest: A Case Report"

_sports, 2023, doi:10.3390/sports11020048_

Round 1

Reviewer 1 Report

Great article on an interesting topic. If space permits, would be nice in the discussion section to address issues related to poor sleep quality and accidents / cognitive function / falls at high altitude, esp. since your case study person had a fall which resulted in an injury that could have jeopardised the mission. 

For example, see a nice review on hypoxia and standing balance here: https://pubmed.ncbi.nlm.nih.gov/33484334/

...and: another here: https://pubmed.ncbi.nlm.nih.gov/31949289/

...and: https://www.liebertpub.com/doi/10.1089/ham.2009.1056

Reviewer 2 Report

Dear authors,
I comment for this field study, not easy to perform. However, collecting data in difficult conditions does not mean that the results are scienfitically interesting and valid.

1) there is no question in this introduction. there is no link with the current physiological and practical knowledge. Hence, the introduction needs to be rewritten so that the reader can understand what you want to do

2) the methodology suffers of several major flaws. Especially, there is no possibility to understand how you compute the HR-derived data. you mentionned several sleep periods during several nights, but in the end: did you average all the sequences? how could you calculate indexes derived from spectral analyses with only 2.5 minutes of data lenght? how have you calculated the correlation? the complete metholodical part should be rewritten so that one can replicate the analyses - this is the basis of science.

3) Discussion: what is the noverlty of your results? what is the meaning of a r=0.4 or 0.31 in the context of your study? it means that there is no real association between the values you collected. so what? In the end, you mentionned that there is a decrease of HR with altitude, and no real link with HRV data - not surprising since the HRV data are not well obtained. Unfortunately, the context of this study does not help in having strong evidences of what happens at very high altitude, from the autonomic nervous system perspective. I don't think your conclusion is supported by your data. Especially: you did not explain how sleep was collected. There is no evaluation of sleep quality (where are the results?) So your conclusion is purely speculative

Round 2

Reviewer 2 Report

Dear authors,

you improve the manuscript but several major flaws still persist, in the same line as in the first round of revision.

There is now a aim with a focus on ANS activity in the introduction. But there is no real added value: why do you expect a difference between below/above 5,500m?

Sympathetic activity: you explicitly mention that LF/HF is influenced by both parasympathetic and sympathetic activity in the method part. So be consistent. Why interpreting this as a sympathetic index only thereafter?

You mention a clinical link with balance while there is no mentionned of this in the Methodology and results parts. This make no sense.

The methodology to statistically analyse the data are still no presented.

The conclusion is not supported by the Method and results: neither on balance nor on well-being. This is overinterpretation of what you have collected/presented and hence is not compatible with acceptance of this manuscript in a scientific journal.
